# Lived experiences of informal caregivers of adolescents with bipolar disorder at Masaka regional referral hospital

Philip Amanyire [1]*, Jane Kasozi Namagga[1], Grace Nambozi[1], Samalie Nakanjako[1‡], Abel Rubega [2‡], Samuel Maling[2]

1 Department of Nursing, Mbarara University of Science and Technology, Mbarara, Uganda,
2 Department of Psychiatry, Mbarara University of Science and Technology, Mbarara, Uganda

☯ These authors contributed equally to this work.
‡ SN and AR also contributed equally to this work.
* philipamanyire@gmail.com, 2022mns017@std.must.ac.ug

## Abstract

Bipolar disorder (BD) among adolescents exacerbates physical, mental, and social challenges, resulting in strained relationships with family and friends. The study aimed to explore the lived experiences of informal caregivers of adolescents with BD at Masaka Regional Referral Hospital. A qualitative phenomenological design, guided by van Manen's approach, was employed to understand the lived experiences of 11 informal caregivers of adolescents with BD. Data were analyzed using van Manen's six-step framework. Five key themes emerged: navigating complexities, hidden struggles of caregiving, caregiver relationships, support networks, and stigma. Informal caregivers experienced struggles, stigma, and a lack of family support, compounded by insufficient psychoeducation about BD. However, the community and mental health workers were very supportive. Thus, there is a need to prioritize the provision of psychoeducation about BD among adolescents and their family members to help improve the family support and involvement in care.

## Introduction

Bipolar disorder (BD) presents significant challenges for adolescents, impacting not only their lives but also those of their informal caregivers [1]. Bipolar disorder is defined as a group of brain disorders that cause extreme fluctuation in a person's mood, energy, and ability to function [2], typically manifesting in late adolescence or early adulthood [3,4]. A recent meta-analysis in Europe indicates that the prevalence of BD among adolescents is increasing [5], with its incidence rising from 84.04 per 100000 individuals in 1990 to 87.60 per 100000 in 2019 [6]. In Uganda, research at the National Psychiatric Referral Hospital indicates that bipolar disorders constitute 30–35% of admissions [7,8], and it is recognized as the most prevalent severe

**Data availability statement:** All relevant data are within the paper and its Supporting Information files.

**Funding:** The authors received no specific funding for this work.

**Competing interests:** The authors have declared that no competing interests exist.

psychiatric disorder, affecting 66.4% of patients [9]. These studies have focused mainly on adults, yet the prevalence of any mental disorder among children under 18 years is estimated at 22.9% [10].

Informal caregiving involves providing unpaid care or support to individuals who cannot care for themselves due to a disability or long-term health condition [11]. This care is often provided by family members or close friends who play a critical role in providing support and stability to adolescents navigating the complicated terrain of BD [12]. These caregivers often have more free time, are the closest relatives, and actively participate in family decision-making processes [13]. Their responsibilities include: ensuring adolescents attend appointments in the clinic, communicating with healthcare providers, occasionally representing their patients at appointments when necessary, and helping with daily tasks such as bathing and house cleaning [14,15]. However, some caregivers struggle to provide adequate care for adolescents with BD, often due to a lack of knowledge about mental health [16,17], which can lead to increased stress levels [18]. The Sub-Saharan Africa region continues to face unique challenges in mental health care, including limited resources and diverse cultural contexts [19]. Informal caregivers in this region, mainly family members, provide care with love, affection, commitment, empathy, compassion, and a genuine willingness to support a mentally ill person [20]. Recognizing the significance of cultural sensitivity in mental health research is essential [21,22], as it highlights the influence of cultural, societal, and familial factors in shaping the lived experiences of informal caregivers.

In Uganda, the role of informal caregivers is recognized as multidimensional, demanding, and constantly evolving. These caregivers often adhere to alternative belief systems, relying heavily on traditional and faith healers, rather than professional mental health services [23]. In addition, a recent scoping review highlights a significant gap in research focusing on caregivers of patients with non-communicable diseases [24]. Studies have documented several challenges faced by informal caregivers, including unreliable access to care, which has financial, psychological, social, and health implications [25], parenting stress is also prevalent due to increased hospitalizations [26]. Additionally, caregivers often lack the time to prioritize their well-being [27,28] and face stigma associated with mental illness [29]. These problems are intensified by interpersonal conflicts and family disruptions among the relatives of adolescents dealing with mental illness [20], creating serious concerns for the affected families [30]. Adolescence is marked by profound physical, mental, and social changes [31]. The onset of Bipolar disorder (BD) during this critical period often results in strained relationships between adolescents and their caregivers [28,29]. Informal caregivers play a critical role in supporting adolescents with BD [12]. However, they often experience stress and emotional difficulties, which can lead to strained marriages, disrupted family relations, workplace challenges, and negative interactions within their communities [1]. There is limited information on the experiences of informal caregivers of adolescents with BD at Masaka Regional Referral Hospital (MRRH), the study site. This study aimed to explore and gain a deeper understanding of the lived experiences of the informal caregivers of adolescents with BD at MRRH.

## Methodology

### Study design and setting

The study employed a cross-sectional qualitative phenomenological design, guided by van Manen's framework [32] to explore and interpret the meaning and essence of everyday lived experiences of informal caregivers. It combined descriptive and interpretive methods to explore, interpret, and critically reflect on the world as a horizon within which all our experiences take shape [32]. The research was carried out at the Psychiatric Unit of Masaka Regional Referral Hospital (MRRH) situated in Masaka city, approximately 132 kilometers (82 miles) southwest of Kampala, Uganda's capital. MRRH serves as a referral centre for psychiatric cases from surrounding districts in the greater Masaka region, including Lyantonde, Sembabule, Kalungu, Lwengo, Bukomansimbi, Kalangala, and Rakai, and Masaka itself.

### Study population

The study population involved Informal caregivers of adolescents with BD receiving treatment from the Psychiatric unit at MRRH for the duration of the data gathering. The adolescents' diagnoses were made by Psychiatric Clinical Officers trained, certified, and licensed by the Uganda Allied Health Professionals Council.

### Sampling and recruitment

Eleven participants were purposively sampled and recruited for the study from 1st July 2024–22nd July 2024. This sample size was determined after reaching thematic redundancy, a point at which no new themes emerged during in-depth interviews with the informal caregivers of adolescents with BD. To achieve redundancy, the researcher began data coding and identified patterns early as data collection continued. The new interviews were observed to see if they were producing novel insights or simply repeating existing themes. Participants were eligible if they were informal caregivers of adolescents with BD receiving treatment at MRRH. They needed to have lived with the adolescents for at least six months. Additionally, they had to be actively involved in daily care and support, providing detailed experiences. The exclusion criteria included caregivers who were not staying with the adolescents most of the time or those who were unwell during the data collection period.

### Data collection procedures

Data were collected using an in-depth interview tool developed based on existing literature on the lived experiences of Informal caregivers. The tool consisted of open-ended questions specifically tailored toward informal caregivers of adolescents with BD. To ensure consistency, the interview tool was translated into Luganda by two experienced mental health research assistants from the Nursing department and then back-translated into English by two other experienced research assistants. At first, they worked independently, but later compared the original and back-translated versions of the research tool. This helped to resolve any discrepancies through discussion and consensus to ensure consistency and proper flow of the tool.

The researcher was introduced to potential participants by the in-charge at the Outpatient Department, where caregivers had accompanied adolescents for medication refills. The study procedures were thoroughly explained to the potential participants, and informed consent was obtained from those who agreed to take part in the study. In line with hermeneutic phenomenological tradition [32], face-to-face, audio-recorded interviews were conducted using open-ended questions. Each interview lasted for 45–60 minutes. A trained research assistant was present during the interviews, and she helped with taking field notes and noting of the non-verbal cues exhibited by participants, while a study counsellor remained on standby in the nearby room, ready to offer support and counselling services to participants exhibiting distress during the interview process. The interview tool was pre-tested with two caregivers of adolescents with BD at Kiyumba HC IV

to refine its usability, but data from these pre-test interviews were not included in the final analysis. Kiyumba HC IV was chosen because it is within the same geographical location as MRRH, and it also offers mental health services. Participants were asked to narrate their lived experiences as informal caregivers of adolescents with BD. To ensure privacy and maintain anonymity, participants were assigned identifiers denoted as "C".

**Data management and quality control.** The audio recordings and transcripts of the participants were stored on a password-protected computer, while the participants' consent forms were kept under lock and key to ensure the safety of the data.

Trustworthiness in this study was based on the model developed by Lincoln and Guba [33]. Credibility was ensured by verifying the data's overall representativeness and member checking to verify data consistency and accuracy [33,34], which helped in refining the transcripts to ensure that they were the true reflections of the participants. The interviews were held in a secluded clinical room free from interruptions. The participants were interviewed upon arrival at the facility before getting engaged in other activities, such as waiting in queues to see the clinician and lining up for medications, which would otherwise make them exhausted. As part of ensuring the researchers' reflectivity, the following was noted: the author is a practicing mental health nurse based at a primary health care facility, with direct experience providing treatment to individuals with mental health disorders. This professional background provided an insider perspective on the challenges faced by informal caregivers of adolescents with BD. While this familiarity offered valuable contextual insight, it also posed potential interpretive biases. To address such bias, peer participation and member checking during data analysis were employed to ensure critical self-awareness and uphold the integrity of the participants' lived experiences.

## Data analysis

The researcher employed a phenomenological approach based on the work of van Manen [32], following six steps: (i) focusing on the nature of lived experience, (ii) exploring experiences as lived, (iii) identifying essential themes, (iv) hermeneutic writing and rewriting, (v) maintaining a strong connection with the phenomenon, and (vi) balancing the whole and its parts. Audio recordings were transcribed verbatim, and the transcripts were analyzed in a repeated, ongoing cycle, which helped in determining thematic redundancy by the 11th participant. To ensure accuracy, the researcher immersed himself in the data, identified emerging themes, and conducted member checking when necessary. This analysis resulted in five themes emerging from fourteen subthemes.

## Ethics statement

Approval to conduct this study was obtained from: Mbarara University of Science and Technology Faculty Research Committee (FRC), Mbarara University of Science and Technology Research Ethics Committee (MUST-2024–1527), and the administration of Masaka Regional Referral Hospital. Participants provided written informed consent before participating in the study, confidentiality was maintained by excluding names of the participants from the transcripts, and privacy was ensured by holding interviews in a private setting.

## Results

Eleven participants were recruited for the study, with their details presented in Table 1.

### Themes and subthemes

Five themes and fourteen subthemes emerged from the participants' narrations of their lived experiences while caring for adolescents with BD. These included navigating complexities, hidden struggles of caregiving, caregiver relationships, support networks, and stigma. Table 2 presents an overview of the emerging themes and subthemes.

**Table 1. Characteristics of the study participants.**

| Participant | Sex | Religion | Marital status | Age (years) | Relationship with the Adolescent |
|---|---|---|---|---|---|
| Caregiver 1 (C1) | Female | Muslim | Separated | 38 | Mother caring for an 18-year-old daughter |
| Caregiver 2 (C2) | Female | Muslim | Married | 36 | Mother caring for a 14-year-old daughter |
| Caregiver 3 (C3) | Female | Muslim | Separated | 57 | Mother caring for a 19-year-old daughter |
| Caregiver 4 (C4) | Female | Anglican | Married | 39 | Mother caring for a 14-year-old daughter |
| Caregiver 5 (C5) | Female | Born-again | Married | 42 | Mother caring for a 17-year-old daughter |
| Caregiver 6 (C6) | Female | Catholic | Widow | 74 | Grandmother caring for an 18-year-old granddaughter |
| Caregiver 7 (C7) | Female | Catholic | Widow | 65 | Grandmother caring for a 17-year-old grandson |
| Caregiver 8 (C8) | Female | Catholic | Separated | 53 | Mother caring for a 19-year-old son |
| Caregiver 9 (C9) | Female | Muslim | Separated | 43 | Mother caring for a 17-year-old son |
| Caregiver 10 (C10) | Female | Catholic | Married | 40 | Mother caring for a 17-year-old son |
| Caregiver 11 (C11) | Female | Catholic | Married | 52 | Auntie caring for a 19-year-old son |

**Table 2. Summary of themes and subthemes.**

| Themes | Subthemes |
|---|---|
| Navigating complexities | Limited understanding of bipolar disorder |
| | Inability to seek care |
| | Worry about the adolescent's future |
| Hidden Struggles of Caregiving | Financial burdens |
| | Resistance |
| | Exhaustion |
| Caregiver relationships | Interaction with the adolescent with BD |
| | Caregiver-family interactions |
| | Caregiver-community relationship |
| Support networks | Sparse family assistance |
| | Community as a pillar of support |
| | Mental health workers |
| Stigma | Fear of gossip and misrepresentation |
| | Affiliate stigma |

**Theme 1: navigating complexities.**

This theme emerged from three subthemes: limited understanding of bipolar disorder, inability to seek care, and worry about the adolescent's future. This highlights the participants' misconceptions about BD.

**Subtheme 1: limited understanding of bipolar disorder.** Most participants attributed BD to witchcraft or evil spirits, while a few mentioned genetics, substance use, or academic pressure. Initial symptoms were often mistaken for typical adolescent behavior or misbehavior, causing delays in the recognition of BD. None of the participants were aware of the adolescent's diagnosis as exemplified in their statements:

*"Some people say that she was bewitched or that these are family spirits that need to be attended to"* [C2]

*"It took me a while to notice, but I knew she was a calm child, when she started talking a lot saying they had night dancers at school I started to worry"* [C6].

*"Up to now, I have not had a chance of getting a health worker informing me about the exact condition that my child is suffering from."* [C5 & C3]

The above experiences ended up causing the caregivers to heavily punish the adolescents with BD due to mistaking of their misbehavior, while in some instances they ended up seeking better mental health care alternatives from the higher-level facilities due to lack of comprehensive information about the adolescents' diagnosis.

*"In fact, at that time I thought to myself that she was being disrespectful and that I needed to correct her immediately by punishing her."* [C12]

*"I didn't get any information and that is how I suggested for her to be taken to Butabika hospital because I thought they had machines to check what was wrong with her head but they also just gave her medicine."* [C6]

**Subtheme 2: inability to seek care.** Most participants faced challenges in accessing formal care and often turned to traditional healers despite the difficulties. One participant, for example, shared a negative experience with a traditional healer.

*"…we spent three months in the shrine and I was sleeping on the rocky ground, we were fetching water from far away, the child used to escape from us and we had to always look for her."* [C6]

For some, prayer was the preferred method, with hospital care considered only as a last resort after other treatment options failed.

*"Me and the patient dedicated ourselves in God's hands and we have been praying, Mark was even serving in the church…"* [C8]

Surprisingly, some participants reported using crude methods, such as locking up or restraining adolescents with BD, to manage their condition was noted by some participants.

**"…we reached a point of restraining her with ropes and she even has scars on her hands."** [C6]

**Subtheme 3: worry about the adolescent's future.** Participants expressed constant worry about the adolescent's future independence. Their worry was based on focusing on the fear of leaving the adolescents alone, and concerns about their anticipated future.

*"…sometimes I am there thinking that maybe after my child growing up, she would go abroad for work but with this illness, I do not think that she can go"* [C1]

*"With these irresponsible men in the village, I fear to leaving her home alone, they can rape her, whom will you ask if such happens?"* [C2]

This led to some participants facing challenges with replacing or paying for the damaged properties as narrated by one of the participants;

*"...what hurt me was destroying my things which I have no hope of replacing soon … as for the clothes he threw in the toilet, the basins he broke, I have never replaced those"* [C7]

**Theme 2: hidden struggles of caregiving**

This theme relates to various struggles that informal caregivers of adolescents with BD go through, such as navigating the financial burden, resistance, and exhaustion.

**Subtheme 1: financial burdens.** Most participants described the high costs associated with caring for adolescents with BD, including medication costs, daily care, and trekking long distances to access health facilities.

*"…the transport, I use 56,000 shillings, when I move with her to the health facility. Sometimes I am forced to leave her at home because of lack of enough transport money." [C2]*

**Subtheme 2: resistance.** Participants described how they encountered resistance from adolescents with BD, often struggling to follow instructions. However, before the onset of BD, these adolescents were cooperative and well-disciplined.

*"When she starts talking or quarreling or abusing, she does not know that this is my parent or this is so and so, she abuses all of you, and I say oh my God!" [C4]*

**Subtheme 3: exhaustion.** All caregivers expressed feeling overwhelmed by the demands of caring for adolescents with BD, describing the experience as difficult, frustrating, and highly challenging.

*"The situation is not easy because whenever I look at her when she is depressed or she has refused to eat, I find it difficult to handle." [C3]*

*"That situation is not easy, it is a difficult one, it requires some patience because such a person most of the time does a lot of annoying things and does them by force and is irritable." [C4]*

**Theme 3: Caregiver relationships**

Participants shared diverse experiences regarding their interactions with adolescents with BD, their families, and the community.

**Subtheme 1: interaction with the adolescent with BD.** Most participants reported having good relationships with the adolescents before the illness. However, the onset of BD caused difficulties, which notably improved with treatment. One caregiver acknowledged that their poor relationship stemmed from how they initially handled the adolescent before recognizing the illness.

*"At first, she used to fear me a lot because at first, I wasn't aware that she was mentally sick so whenever she made a mistake, I would punish her a lot." [C2]*

**Subtheme 2: Caregiver-family interactions.** Some participants experienced poor relations with their families while others maintained good relationships as in the following quotes:

*"The relationship was not bad because they all used to help me and they used to come wherever they were to come and visit me, some we used to come with them to the hospital, they are the ones who used to catch him." [C2]*

*"They do not care about me, their home is not far from here, I took her there, they just said sorry to me and they told me to be strong as a woman, I did not receive any support from them." [C1]*

**Subtheme 3: Caregiver-community relationship.** Most participants reported positive interactions with community members, who were generally very supportive.

*"I have good relationships with the community members, my son's illness has not changed anything because, they talk well to him and relate with him well, he has never destroyed people's property or abused a neighbor's child, or beating them." [C8]*

**"Community members where I stay, they love me and Junior so much."** [C9]

**Theme 4: support networks**

Participants expressed mixed reactions to the support they received from family members, community members, and mental health workers.

**Subtheme 1: sparse family assistance.** At the family level, eight participants indicated that they did not receive support from their close relatives, for example, participant 8 noted that;

*"I do not have support from family…it's me who takes care of him, when money gets lost, he does not get medicine."* [C8]

Another one noted the importance of collective effort during the challenging times of the adolescent's breakdown as acknowledged in the quote below;

*"…if you work together, one with 5,000 shs and sends it, one says let me sell my coffee and support you - we get challenges but if you work together you win."* [C11]

**Subtheme 2: community as a pillar of support.** All the participants frequently narrated how they received support from the community members. For example, one of them narrated how the community helped her get her daughter;

*"She was in a taxi, and the taxi people called me, suspecting that she had become mentally sick, and said they would not let her go out of the taxi before I reached there to receive her, that is what they did."* [C3]

**Subtheme 3: mental health workers.** Most participants had positive experiences with mental health workers, reporting good interactions during the admission period, and follow-up reviews. However, they were not satisfied with the mental health workers' failure to provide in-depth information regarding the causes, diagnosis, and medications for their patients.

*"Truthfully, he did not tell me the outcome of the interview, he did not inform me about the cause of my child's mental health problem."* [C2]

**Theme 5: stigma**

Some participants experienced stigma, which manifested mainly in two forms: fear of gossip and misrepresentation, and affiliate stigma.

**Subtheme 1: fear of gossip and misrepresentation.** Six of the eleven participants expressed the fear of gossip and misrepresentation of their adolescents' behavior during a bipolar disorder episode. For example, participant C5 narrated that;

*"…there is a neighbor who visited us and when she went back, she informed those people that Rebecca was throwing stones at me, something which wasn't true."* [C5]

**Subtheme 2: affiliate stigma.** Some of the participants experienced affiliate stigma, they were reluctant to share their adolescent's condition with others, such as extended family members or community members, as narrated by C4;

*"I do not want people to know that my daughter has this kind of illness, because I know that if anyone gets to know about it, many people say a lot of things"* [C4]

## Discussion

Informal caregivers of adolescents with BD expressed various lived experiences with the adolescents. They experienced exhaustion from constant support, managing mood swings, and ensuring treatment adherence. Behavioral resistance

complicates caregiving, as many recalled a time when the adolescent was more cooperative. To better interpret the results, responses were categorized into specific themes.

### Navigating complexities

Most participants attributed bipolar disorder (BD) to witchcraft or evil spirits, reflecting cultural beliefs and a lack of understanding of its biological and psychological causes. These attributions further reveal caregivers' culturally grounded efforts to make sense of their adolescents' conditions in a world disrupted by mental illness. A few participants, however, recognized factors like genetics, substance use, or academic pressure. The belief in these supernatural causes leads to caregivers pursuing traditional treatments such as traditional healers and witch doctors, hence delaying the utilization of evidence-based therapies, which was well noted during the interviews, in which hospital treatment was a last resort after failing with other treatment options. These findings align with [35], who noted similar beliefs about witchcraft and evil spirits causing mental illness, and with [17], who observed a poor understanding of mental illness among caregivers. This diversity in views underscores the need for mental health education to bridge traditional beliefs with modern science, helping caregivers and the community manage BD more effectively while reducing stigma and improving support for affected adolescents.

Despite all participants effectively identifying their patients' symptoms, some caregivers struggled to distinguish between typical and atypical adolescent behavior. Such confusion and delayed recognition of signs and symptoms of BD point to a fundamental breakdown in the meaningful orientation of the caregivers within their caregiving role. This echoes Van Manen's notion of the fragility of meaning structures when the familiar becomes estranged. This often resulted in caregivers administering harsh punishments, which hindered the early detection of bipolar disorder in these adolescents. This confusion may stem from the rapid physical, emotional, and social changes typical of adolescence. These findings align with [18], who noted that participants could recognize these symptoms, while [36] found that caregivers often misunderstood the signs and symptoms until after a diagnosis. The caregivers' lack of diagnosis-specific information from mental health workers further prevented them from understanding the condition, leaving many unaware of the clinical name or nature of BD. This absence of clarity deepened their existential uncertainty and pushed some to seek diagnostic assurance from higher-level facilities. This sheds light on Van Manen's form of existential orientation in which people tend to reclaim meaning amidst a state of ambiguity. These findings are in line with [16,18], who documented a poor understanding of the mental illnesses by caregivers, which reduced their involvement in care. Medical treatment was a last resort for many of the participants, with many of them first seeking treatment from traditional healers, herbalists, and religious people, which ended up leading to inconsistencies in care. Some caregivers' narratives of sleeping in shrines or physically restraining adolescents reflect their efforts to maintain control and offer protection, even under unusual circumstances. This reflects van Manen's concept of embodied relationality, which interprets such actions not as failures but as deeply situated moral responses to suffering. The caregivers ended up combining different forms of treatment for the sick adolescents, and this is a common practice [31,36]. In addition, a strong reliance on traditional and faith healers has been noted in Uganda [19,37].

All participants expressed worry about the independence of the adolescent with BD, they feared that such an adolescent could harm others or destroy property, or be raped in the absence of the parent. As such, they were prompted to constantly monitor such adolescents, which turned out to be tedious and exhausting. To the caregivers, they seemed to experience the adolescents' future as foreboding - the future that appears not as a promise but as uncertainty and potential loss. Their narratives reveal a deep tension between hope and fear, shaped by their relational commitments and social expectations. This is in alignment with [38], who noted that caregivers view people with mental illness as dangerous. Concerns about the adolescents' future independence, including employment, marriage, and health, without ongoing caregiver support, were narrated by most of the participants, which led to high stress and overly protective behavior. These concerns align with other research showing caregivers' constant anxiety about their patients' future and their dependence [38,39].

### Hidden struggles of caregiving

The participants experienced the financial burden while caring for adolescents with BD. They faced high costs for transportation, medication, and traditional healer consultations, leading to treatment lapses, relapses, and debt. This subtheme leads to disruption of relational meaning as per van Manen, which makes it hard for caregivers to decide on which treatments to prioritize and how far to travel, thus interrupting their ability and capacity to act. Mental health workers should advocate for increased funding and resources for families caring for adolescents with BD. This is in line with other researchers who noted the same challenges [40–42], underscoring the need for financial support [43]. Almost all participants reported a difficult shift in the behavior of the adolescent with BD from previously cooperative and disciplined to undisciplined and resistant behavior, reflecting common symptoms of the disorder, such as dramatic mood and energy changes [2]. The statement *"she does not know that this is my parent…"* as narrated by one of the caregivers reveals a profound break in relational continuity. Caregivers perceive this behavior not just as defiance, but as a break in their relationship and the adolescent's identity, as the adolescent no longer reflects their shared past. This abrupt change can be shocking and stressful for the informal caregivers, aligning with [43], who found managing BD challenging due to unpredictable symptoms. They also felt exhausted from handling the adolescents' demands alone, highlighting the need for respite care and self-care. This exhaustion can be interpreted not just as tiredness, but as a phenomenological condition of being stretched beyond one's limits. Their narrations about exhaustion are lived expressions of the caregiver's diminishing capacity to remain grounded in the face of constant challenge. Positive support came from observing other caregivers managing similar challenges. Issues like insufficient focus on caregiver well-being [23,24,44] and stigma associated with mental illness [29] contribute to caregiver burden, with some viewing their patients as a burden [45].

### Caregiver relationships

Participants frequently reported strained relationships with adolescents after the onset of illness. Their narratives reflect a shift from close, affectionate relationships with the adolescent before illness onset to strained and challenging dynamics during episodes of mental illness. Bipolar disorder disrupts the caregiver's sense of relational continuity and recognition - a shared history and intimacy that once defined the relationship. However, as treatment and recovery progressed, the fractured bond would be repaired, indicating how the illness had heavily impacted their relationship. The mood swings, behavioral changes, and symptoms of BD likely contributed to these challenges. These findings align with [46], who found that mood symptoms in adolescents with mental health issues strained their relationships. Such mood changes, ranging from euphoric to depressed behaviors [2], are also linked to increased caregiver stress [26]. Most of the study participants reported strained relationships with family members but had positive interactions with the community members. This is consistent with [16], who observed that family members often exclude individuals with mental illness from important decisions, such as those involving family inheritances, and may even be a source of discrimination. The positive interactions with community members provide emotional relief, practical assistance, and a sense of social inclusion. Such dynamics underscore the important role that the broader community plays in alleviating caregiver burden and enhancing resilience. Thus, there is a pressing need to more effectively integrate community resources into formal mental health care systems, particularly in low-resource settings like Uganda. The community should be recognized as a legitimate and valuable partner in caregiving. Highlighting the need of including community-based support mechanisms within the national and district-level mental health policy frameworks.

### Support networks

The study revealed that while community members were highly supportive of informal caregivers of adolescents with BD, most family members were not. Many participants expressed disappointment and burden due to limited or non-existent support from family members. This absence was experienced not just as a logistical hardship but as a deep emotional disconnection. This lack of family support may lead to burnout and emotional distress of the caregiver, as family is expected

to be the immediate pillar of support. These findings align with [47], who noted that caregivers often lacked support from both family and others, particularly as the patient's symptoms worsened. The community's positive support role acts as a protective container, affirming that caregiving is not a private struggle but a shared social duty. Additionally, while participants appreciated the care from mental health workers during hospitalization, they were dissatisfied with the lack of communication regarding causes, diagnoses, and medications. Echoes such as *"He did not inform me about the cause of my child's mental health problem"* reflect a rupture in trust and epistemic access, where the caregiver, already shouldering intense emotional and physical labour, is left disempowered by partial silence in the clinical encounter. While mental health workers were physically present, their relational absence in terms of dialogue and education led to an ambiguous support experience for the caregivers. This led some caregivers to transfer patients to higher-level facilities like Butabika National Referral Hospital. The findings echo [43], who reported caregiver frustration with the healthcare system due to issues like medication, poor treatment, and misdiagnoses. This highlights the need for better education and communication from mental health providers to support caregivers effectively, which is crucial for treatment implementation [48].

### Stigma

For many participants, the most wounding form of stigma came from being misrepresented or misunderstood by others, such as extended family members or community members. The caregiver's narratives speak volumes about the caregiver's vulnerability to social judgment, where the adolescent's behavior is interpreted through a distorted lens, often stripped of medical or compassionate context. In this gaze, both the adolescent and the caregiver are alienated, no longer seen as subjects with meaning but as objects of fear or ridicule. This may be due to the unpredictable nature of bipolar disorder, which results in behaviors that are difficult to explain to those who do not understand them. This is in line with [49] who identified caregivers of bipolar disorder patients as among the most stigmatized, underscoring the importance of family support in caregiving.

### Limitations

The study excluded Adolescents and other stakeholders, yet incorporating their views could have offered a more holistic understanding of the caregiving context.

The study highlights the lived experiences of only female Informal caregivers, and thus it lacks the voices from the male counterparts who could have enriched the discussion.

The study findings reflect the lived experiences of participants whose adolescents with BD were already receiving mental health care at the facility. These results may not represent those who stay at home or seek treatment from traditional healers.

### Conclusion

Qualitative interviews revealed that informal caregivers experienced struggles, stigma, and a lack of family support, coupled with insufficient psychoeducation from mental health workers. However, they received support from the community members and mental health workers. Despite these challenges, caregivers remain resilient, highlighting the need for family support, financial aid, mental health education, and accessible healthcare.

Thus, the Mental health workers should prioritize the provision of psychoeducation about BD among adolescents to help informal caregivers understand and adhere to appropriate treatment plans.

### Supporting information

**S1 File. A PDF document file containing transcripts of the study participants' detailed descriptions of their lived experiences.** It includes eleven transcripts labelled from C1 to C11.
(PDF)

## Acknowledgments

We acknowledge the administration of Masaka Regional Referral Hospital for the support throughout the data collection period, and the contribution of the Informal caregivers of Adolescents with BD who constituted our study sample.

## Author contributions

**Conceptualization:** Philip Amanyire.

**Data curation:** Philip Amanyire, Jane Kasozi Namagga, Grace Nambozi, Abel Rubega, Samuel Maling.

**Formal analysis:** Philip Amanyire, Jane Kasozi Namagga.

**Investigation:** Philip Amanyire.

**Methodology:** Philip Amanyire, Jane Kasozi Namagga.

**Project administration:** Philip Amanyire.

**Resources:** Philip Amanyire.

**Supervision:** Philip Amanyire.

**Validation:** Philip Amanyire.

**Visualization:** Philip Amanyire, Grace Nambozi, Abel Rubega, Samuel Maling.

**Writing – original draft:** Philip Amanyire.

**Writing – review & editing:** Philip Amanyire, Jane Kasozi Namagga, Grace Nambozi, Samalie Nakanjako, Abel Rubega, Samuel Maling.

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
