## [Decision Letter · Decision Letter 0]

PMEN-D-25-00087

Lived experiences of Informal caregivers of Adolescents with Bipolar disorder at Masaka Regional Referral Hospital

PLOS Mental Health

Dear Dr. Amanyire,

Thank you for submitting your manuscript to PLOS Mental Health. After careful consideration, we feel that it has merit but does not fully meet PLOS Mental Health’s publication criteria as it currently stands. Therefore, we invite you to submit a revised version of the manuscript that addresses the points raised during the review process.

We look forward to receiving your revised manuscript.

Kind regards,

Jonathan Han Loong Kuek, Ph.D.

Academic Editor

PLOS Mental Health

Journal Requirements:

Reviewers' comments:

Reviewer #1: General comments and some key concerns:

1. It is an interesting study that is giving an insight on the “Lived experiences of Informal caregivers of Adolescents with Bipolar disorder at Masaka Regional Referral Hospital”. However, there are some issues that need to be addressed as indicated.

2. Abstract

• The authors should follow the journal guideline format in writing the abstract

• The authors need to summarize the key finding on the lived experiences of Informal caregivers of Adolescents with Bipolar disorder. Some of the findings in the conclusion should have been in the results section.

• In the conclusion, what are the key issues that can be concluded from the finding of the study

3. Introduction

• Line 43: The authors should indicate where the prevalence of Bipolar disorder is increasing

• Line 45: The authors state that the prevalence of Bipolar disorder is scarce in Uganda, however, there many publications in this area in Uganda, and even in Butabika National referral hospital - Studies at the Butabika National Psychiatric Referral Hospital indicate that bipolar disorder is the most common diagnosis among patients seeking treatment, with estimates of up to 66.4% of cases (see. https://ibpf.org/wp-content/uploads/healthy-living-book/Bipolar-Disorder-in-Uganda.pdf).

4. Methods

• In this sub-section, the authors should include the following sub-section; study population or unit

• Line 119: The authors should replace “The interview guide” with “interview tool”; and the authors should explain why the pre-tested with two caregivers of adolescents with BD were done at Kiyumba HC IV.

• How did the caregivers know that it was Bipolar disorder and not any other disorder?

• The authors should explain how the data managed and quality control was achieved

• Line 124: The authors should write data analysis

5. Results

• Line 141: 3. “Findings” should be “Results”

• The authors should elaborate more on the demographic characteristics of the study participants including level of education, occupation if possible, religion, type of family since these can influence the outcomes!!

• What is the unit of age in Table 1 – Line 143?

• Were there BD patients on the treatment and given that it is almost chronic disease and therefore requires prolonged treatment, did the complexity of treatment and adherence not an issue!!!

6. Discussion

• Line 303: what are the traditional treatments, these individuals seek for?

• The limitations should be based on methodological utilized to answer the objectives of the study and not as it is stated

7. Conclusion

• Recommendation should be combined in the conclusion

8. References

• The text should be uniform all through and not a mixture of upper case and lower case. See the Journal guideline

Reviewer #2: Dear authors

Thank you so much for taking the opportunity to review this paper. Here are my comments:

Title

Use title case for academic consistency (capitalize major words).

Avoid line breaks mid-title (unless it's a formatting constraint in the journal style)

Abstract

Replaced casual/ambiguous phrases like "circle of support" with more precise terms like "support networks."

Changed "struggles" to "emotional burden" for better academic tone.

Improved flow by reducing passive constructions and redundancy (e.g., "coupled with" → "compounded by").

Introduction

Replaced semi-colons inappropriately used in lists with commas.

Grouped sequential references (e.g., 13–15) for readability.

Clarified the flow between citations and statements.

Replaced casual language with more formal equivalents (e.g., "play a pivotal role" → "play a critical role").

For enhance the quality of your article please use these references:

https://www.jpcmed.com/article_153234.html

https://brieflands.com/articles/ijpbs-123998.html

Reordered some content (e.g., defining informal caregiving earlier) for better narrative structure.

Methodology

Reworded “to gain deeper insights” → “to explore and interpret” for more formal and objective tone.

“Utilized” replaced with “combined” for clarity and conciseness.

• Made the connection between van Manen’s philosophy and the study’s goals clearer.

• Provided a more natural explanation of the phrase “world as world” by integrating it with its philosophical context.

• Smoothed transitions between study design and setting.

• Grouped related geographic information for easier readability.

• Replaced casual phrasing like “had accompanied adolescents for drug refills” with more formal alternatives like “for medication refills.”

• Changed “specifically directed toward” to “specifically tailored to” for a smoother tone.

• Added punctuation and corrected minor grammar issues (e.g., spacing, sentence fragments).

• Clarified the role of the assistant and counselor to avoid ambiguity.

• Broke long sentences into clearer components.

• Ordered information for a natural progression: tool development → participant recruitment → ethical consent → interview conduct → safety considerations → confidentiality.

Reviewer #3: Thank you for the opportunity to review this thoughtful and socially relevant manuscript. Your study contributes meaningfully to the limited body of literature on the lived experiences of caregivers of adolescents with bipolar disorder in low-resource settings. The cultural and emotional complexity of caregiving is well captured, and the participant quotations enrich the narrative, giving voice to often unheard experiences.

I offer the following suggestions for improvement:

Major Issues:

1-Theoretical and Conceptual Framing :The manuscript would benefit from a stronger theoretical foundation. While van Manen’s phenomenological framework is mentioned, it is not sufficiently discussed in terms of how it informed data interpretation and writing. Consider elaborating on how this framework shaped your lens and analytic decisions.

2-Methodological Rigor: The claim of data saturation is not substantiated. Please clarify how saturation was determined (e.g., thematic redundancy, use of saturation grids).

Describe how trustworthiness was ensured, did you use triangulation, member checks, audit trails, or inter-coder reliability?

The translation/back-translation process needs more detail. How was accuracy ensured? Were transcripts validated?

3-Researcher Reflexivity Phenomenological research requires a discussion of the researchers' positionality and potential influence on data interpretation. This is missing and should be added.

4-Sample Representation: All caregivers were female (mothers, grandmothers, one aunt). While understandable, this should be acknowledged as a limitation, and the absence of male voices discussed.

5-Depth of Analysis in Discussion: The discussion section largely restates results. Consider expanding the analysis by connecting findings to broader regional/global literature, including caregiver burden models, cultural stigma, and task-sharing in low-resource mental health settings. Please discuss how community support might be better integrated into formal care systems or leveraged for policy development.

Minor Suggestions:

1-Please proofread the manuscript for grammatical errors and clarity. Some sentences (e.g., in the discussion) are long or awkwardly phrased.

2-Tables need better formatting and clearer headings.

3-In the abstract and introduction, please specify more clearly that this is a Ugandan setting and define BD earlier in the abstract.

4- Please expand slightly on ethical safeguards (e.g., data storage procedures) for completeness.

---

## [Editor Report · Decision Letter 1]

Lived Experiences of Informal Caregivers of Adolescents with Bipolar Disorder at Masaka Regional Referral Hospital

PMEN-D-25-00087R1

Dear Mr. Amanyire,

We are pleased to inform you that your manuscript 'Lived Experiences of Informal Caregivers of Adolescents with Bipolar Disorder at Masaka Regional Referral Hospital' has been provisionally accepted for publication in PLOS Mental Health.

Best regards,

Jonathan Han Loong Kuek, Ph.D.

Academic Editor

PLOS Mental Health
